# Synthesis of Eco-Friendly Biopolymer, Alginate-Chitosan Composite to Adsorb the Heavy Metals, Cd(II) and Pb(II) from Contaminated Effluents

**DOI:** 10.3390/ma14092189

**Published:** 2021-04-24

**Authors:** Mohammed F. Hamza, Nora A. Hamad, Doaa M. Hamad, Mahmoud S. Khalafalla, Adel A.-H. Abdel-Rahman, Ibrahim F. Zeid, Yuezhou Wei, Mahmoud M. Hessien, Amr Fouda, Waheed M. Salem

**Affiliations:** 1Guangxi Key Laboratory of Processing for Non-ferrous Metals and Featured Data, School of Resources, Environment and Data, Guangxi University, Nanning 530004, China; 2Nuclear Materials Authority, P.O. Box. 530, El-Maadi, Cairo 11381, Egypt; mahmoudsayed24@yahoo.com; 3Department of Organic Chemistry, Faculty of Science, Menoufia University, Shebine El-Koam 00123, Egypt; nhamad059@gmail.com (N.A.H.); dodyhamad95@gmail.com (D.M.H.); adelnassar63@yahoo.com (A.A.-H.A.-R.); ibrahimzeid296@gmail.com (I.F.Z.); 4School of Nuclear Science and Engineering, Shanghai Jiao Tong University, Shanghai 200240, China; 5Department of Chemistry, College of Science, Taif University, P.O. Box. 11099, Taif 21974, Saudi Arabia; m.hessien@tu.edu.sa; 6Botany and Microbiology Department, Faculty of Science, AL-Azhar University, Cairo 11884, Egypt; amr_fh83@azhar.edu.eg; 7Medical Laboratories Department, Faculty of Applied Health Science technology, Menoufia University, Shebine El-Koam 32511, Egypt; waheedsalem1979@gmail.com

**Keywords:** eco-friendly sorbent, cost-effective biopolymers, cadmium and lead contamination, contaminated water treatment

## Abstract

Efficient removal of Cd(II) and Pb(II) from contaminated water is considered a fundamental point of view. Synthetic hydrogel biopolymers based on chitosan and alginate (cost-effective and eco-friendly) were successfully designed and characterized by highly efficient removal contaminants. The sorbents are characterized by FTIR, SEM-EDX, TGA, XPS analyses and textural properties which are qualified by N_2_ adsorption. The sorption properties are firstly investigated by the effect of pH, sorption isotherms, uptake kinetics, and selectivity from multi-metal solution with equi-molar concentration. The sorbent with 1:3 ratios (of chitosan and alginate respectively) is the most effective for metal removal (i.e., 0.81 mmol Cd g^−1^ and 0.41 mmol Pb g^−1^). Langmuir and Sip’s models fitted better the adsorption isotherms compared to the Freundlich model. Uptake kinetics was well fitted by pseudo-first-order rate equation, while the saturation was achieved within 40 min. The sorbent shows good reproducibility through duplicate the experiments with negligible decreasing efficiency (>2.5%). The sorbent was applied for water treatment on samples collected from the industrial area (i.e., 653 and 203 times over the MCL for Cd(II) and Pb(II) respectively according to WHO). The concentration of Cd and Pb was drastically decreased in the effluents as pH increased with removal efficiency up to 99% for both elements at pH 5.8 and SD equivalent 1 g L^−1^ for 5 h.

## 1. Introduction

Metal ion removal from either industrial effluents or wastewater is considered a critical topic according to international regulations and governments [1]. The importance of this point is due to the damages of the biotope and health threats that are affected by the discharge of the hazardous metals. The activities in the mining processes and other industries generate contaminations to soil and water for the surrounding areas. The metal mobility through discharging of the industrial tailing data, leaching (in escaping the filtrates from the hydrometallurgy processes), or flying dust, etc., are the main source of pollution and have a bad impact on human health [2]. Metal contamination of water bodies may naturally occur (i.e., rain and floods) or by anthropogenic effects (i.e., metallurgical activities) [3].

The bioaccumulation of the contaminants for the food chain, persistence, and toxicity, may explain the high attention paid to the evaluation in ground and drinking waters. Organizations have designed strict guidelines for the maximum levels of the contaminants (MCL) for livestock and irrigation of water (i.e., World Health Organization (WHO), European Union (EU), U.S. Environmental Protection Agency (U.S.-EPA), and Food and Agriculture Organization of United Nations (FAO) [1,4,5,6], as reported in the Appendix A (see Appendix A).

Cadmium and lead ions are taken directly by a human from the metal-emitting industries or reached indirectly through other live organisms like plants (i.e., seeds and fruits) which allow the passing of these ions from roots (up to 90% from the soil and 10% form air) [7,8], accumulating in fatty tissues and animal milk. Therefore, humans can easily be exposed to metal poisoning from plants or animals. Around 98% of the lead or cadmium that is found in the atmosphere is produced from human activities. Cadmium and Pb cause anemia, affect the central nervous system [9,10], cause liver damage, structure kidney damage, bone tissue that affects calcium metabolism, and decrease the birth weight [11,12,13,14,15].

Several processes are used for the extraction of metal ions. Precipitation [16,17] and solvent extraction [18,19,20,21,22,23,24] are applied for high concentrate solutions. Adsorption/ion exchange [25,26,27] for a low concentrated solution, along with other common processes were used as membrane technologies [28], and electrolytic techniques [29]. Bio-sorption and biosorbents are easily functionalized by substitute reactive groups for increasing the loading capacity, improving kinetics, and for more efficient recovery [30,31,32]. The presence of amine and hydroxyl groups in the structure of these biopolymers facilitates the chemical modification, while physical designing can be performed by dissolving and shaping beads, fibers, and hollow fibers.

Chitosan (partially deacetylated chitin) is considered as one of the most abundant biopolymers. Its properties are gained through the presence of amines and hydroxyl groups in the polysaccharide moieties which are responsible for the hydrophilicity nature. Protonation of amines in acidic medium gives possible solubility of the solids and makes them easy to shape [33,34]. Coated chitosan with nano or microparticles of magnetite was achieved and documented. Grafting of amines and amino acid moieties for improving the efficient sorption, improving kinetics, selectivity, and stability was established [35,36,37,38,39].

Biosorbents based on algal biomass have been used in the last few decades for recovering rare metals and removal of hazardous ions. Carboxylic groups from alginate, sulfonic groups in fucoidan, and amino groups in proteins are the most important groups in these biopolymers. Conditioning as beads and foams for applications either in a column or through batch process receive great attention in metal extraction [40,41,42,43].

Designing a new kind of hydrogel by different ratios of chitosan (C) and alginate (A) (i.e., 1:2, 1:3, and 1:4 for C and A respectively) was performed. This composite was produced by ionotropic gelation using CaCl_2_ in the presence of glutaraldehyde (GA) as a crosslinking agent (colored part in Scheme 1). The produced sorbents were fully characterized (FTIR, SEM-EDX, TGA, Textural properties (N_2_ adsorption), and XPS). The sorption was tested toward Cd(II) and Pb(II) before being treated on the contaminated water sample. The sorption characterization was considered by the effect of pH, kinetics, isotherms, recycling through series of sorption desorption cycles, selectivity in the presence of associated elements and finally was tested for decontamination of water. All sorption investigations were performed twice, and the average was calculated; the deviation from each experiment did not exceed 2.5%.

## 2. Data and Methods

### 2.1. Data

All chemicals are fine products. Lead (II) chloride (98%) and cadmium (II) chloride (anhydrous, 99.999%) salts (i.e., CdCl_2_ and PbCl_2_, for the synthetic solution), Chitosan (with 90.5% deacetylation degree), glutaraldehyde (50 wt.% in H_2_O), and alginate were supplied from Sigma-Aldrich, (Merck, Darmstadt, Germany). Magnesium chloride hexahydrate (≥98.0%), sodium chloride (≥99.5%), and aluminum chloride anhydrous (99.999%) were purchased from Guangdong Sci-Tech Co., Ltd., (Guangzhou, China). A stock solution of 1000 ppm was prepared for each element and a freshly diluted solution was prepared by deionized water to the desired concentration of the experiment. All other reagents are the Prolabo products—Morillons, France.

### 2.2. Synthesis of the Sorbents

Different ratios of chitosan/alginate were prepared by the following method—one gram of chitosan was dissolved in 25 mL of 1% (W/V) acetic acid solution (prepared three separate times in different flasks). Addition of 2, 3, and 4 g of alginate powder to the chitosan solution with continuous stirring till dissolve (for 1:2, 1:3, and 1:4 of Chitosan: Alginate respectively, so-called CA#2, CA#3, and CA#4). The prepared solutions were kept at room temperature for 7 days, then the solution was added drop-wise to 500 mL of (1%, W/V) CaCl_2_ with 5 mL Glutaraldehyde (50%, W/W) for ionotropic gelation and crosslinking of the alginate and chitosan. The precipitated hydrogel was subjected to stirring for 24 h at 30 °C before decantation and washed several times with ethanol/water then dried at 50 °C for 10 h. The expected structure of the functionalized sorbent was designed in Scheme 1 with the proposed synthetic steps.

### 2.3. Characterization

Several tools were used for specifying the functional groups as well as elucidating the chemical structure. FT-IR spectrometry of dried samples after grinding with (1%, W/W) anhydrous KBr was performed using Shimadzu IRTracer-100 (Shimadzu, Tokyo, Japan). Carbon, N, O, and H were measured quantitatively by an element analyzer (EA) (2400 Series II Perkin-Elmer, Waltham, MA, USA). The morphology of the surface was characterized by a scanning electron microscope (SEM) and semi-quantitative surface analyses were performed on (Phenom ProX, scanning electron microscope (SEM), Thermo Fisher Scientific, Eindhoven, The Netherlands), while the elemental composition of the structure was chemically investigated using energy dispersive X-ray (EDX). Thermal decomposition of the prepared data was performed on TGA-DTA under nitrogen atmosphere using (Netzsch STA 449 F3 Jupiter, NETZSCH-Gerätebau HGmbh, Selb, Germany), the temperature ramp: 10 °C/min, under nitrogen atmosphere. Textural properties (N_2_ adsorption) were used by Micromeritics TriStarII (Norcross, GA, USA) at 77 K, the sample was firstly degassed in nitrogen gas for 5 h at 100 °C. The pH-drift method [44] was used for pH_PZC_ investigation (corresponding to pH_0_ = pH_eq_), a 50 mg of sorbent stirred with 30 mL of 0.1 M NaCl solution for 48 h at different pH values (pH_0_ (initial pH) = 1–14 values). ESCALAB 250XI^+^ instrument (Thermo Fischer Scientific, Inc., Waltham, MA, USA) instrument used for the XPS spectra connected with monochromatic X-ray Al Kα radiation (1486.6 eV). The pressure was adjusted to 10^−8^ mbar., while the energy calibrated with Ag3d_5/2_ and C 1s signals at ∆BE: 0.45 eV and 0.82 eV) respectively. in which the full and narrow-spectrum pass energies were 50 eV and 20 eV, respectively. The pH (at initial or after equilibrium) was measured by S220 Seven Compact pH/Ionometer.

### 2.4. Sorption Tests

Captions of the tables and figures were systematic including the specific conditions. The charts show the mean average after duplicating each experiment twice. The overall deviation of each experiment does not exceed 2.5%, indicating the real reproducibility of the experiments. A fixed amount of sorbent (mg) was mixed with a specific volume of bearing metal solution. Solutions of 0.1/1 N HCl or NaOH were used for controlling the pH values for Pb(II) and Cd(II) also for the equimolar solutions. The pH value is not fixed during the adsorption process that record at the end of the experiment. Samples were collected from the loading experiments were firstly filtered using a filter membrane with pore size 1.1 µm, before measuring by ICP-AES (JY Activa M, Horiba/Jobin-Yvon, Longjumeau, France). Sorption isotherms were performed using the fixed weight of sorbent (m, g), contacted with the solution (V, L) of different concentrations (10:500 mg L^−1^) for 48 h. The sorbent dosage for pH experiments, isotherms, selectivity, and recycling was set to 1g L^−1^ while fixed to 0.3 g L^−1^ for the kinetics experiments. Sorption capacity q_eq_ (mmol g^−1^) and removing efficiency (R%) in the natural sample experiments were determined using the mass balance equations q_eq_ = (C_0_ − C_eq_) × V/m, and R% = (C_0_− C_eq_)/C_0_ × 100, respectively, where C_0_ and C_eq_ (mmol L^−1^) are the metal ions concentrations initially and at equilibrium, V and m are the volumes of the solution (L) and the mass of the sorbent (mg) respectively. The temperature of sorption experiments, pH, uptake kinetics, sorption isotherms were performed at room temperature (22 (±2) °C).

### 2.5. Describing of Sorption Isotherms and Uptake Kinetics

The sorption isotherms and uptake kinetics in this study were modeled using the pseudo-first (PFORE) and the pseudo-second-order rate equations (PSORE [45]) for sorption kinetics. The Freundlich, Langmuir, and Sips [46,47] models were used for the sorption isotherms. Appendix A (see Appendix A) report the parameters of the equation.

### 2.6. Treatment of Real Metal-Containing Groundwater

Nile Delta is considered one of the most populated areas in the world, it contains more than 50 million people [48]. The collected contaminated water sample from Abu Zaabal Lake was enriched with Cd and Pb with a concentration of 1.96 mg Cd L^−1^ and 2.03 mg Pb L^−1^. These concentrations were higher than the maximum contaminant levels (MCL) by 653 and 203 times respectively, Hg and Al also increased by 415 and 629 times respectively than the MCL, while Cu, Zn, and Ni increased with little extend (i.e., 3.2, 1.47 and 31.143 times respectively). The water depth of the studied sample ranged from around 3 m to 16 m. The high salinity (ranged around 3905 mg L^−1^) is due to the dissolved sodium salts of either chloride and sulfate [49,50]. High contamination was reached by sewage wastewater. Besides this, the Abu Zaabal area is famous for several industries such as cement factories, chemicals, and fertilizers factories which assist increasing the possibility of pollution level in the adjacent water, soil, and air environments. The sorption was investigated at 3 different pH values, i.e., 5.8 (original pH of the taken sample), 4 (the optimum pH for sorption from the synthetic solution), and pH 2 (acidic pH). The batch method was used for 5 h of contact, while SD was fixed in all experiments at 1 g L^−1^. The MCL was detected for the studied sample comparing to what was reported by the organizations. This was achieved by analyzing the metal ions in the effluents after sorption; Appendix A shows the composition of interesting elements in the water sample and the relation with the MCL.

## 3. Results and Discussion

### 3.1. Characterization

#### 3.1.1. Fourier Transform Infrared Analysis

Figure 1A shows the FTIR spectra of the prepared sorbents at different ratios of chitosan and alginate (CA#2, CA#3, and CA#4). Figure 1B shows the characterization of CA#3 (the most effective sorbent), after metal sorption and after desorption for five cycles. On the other hand, Appendix A (see Appendix A) shows the sorption characterization of CA#2 and CA#4, after metal sorption and after five cycles of sorption desorption.

From these figures, a series of peaks with different resolve efficiency was observed, confirming the chemical modification and the effectiveness of additives. Peaks at 1035 cm^−1^ and 1093 cm^−1^ in CA#2, 1035 cm^−1^, and 1060 cm^−1^ in CA#3 and strong resolved band at 1022 cm^−1^ for CA#4 are assigned to C (-C,-O, and -N) stretch [51,52]. Different intensities (broad to sharp) of the peaks at 1384 (for CA#2 and CA#3) and 1428 cm^−1^ are attributed to carboxylate salt of alginate. The intensities and broadness of these peaks are related to the alginate ratio, indicating the incorporation of COO^−^ for binding with chitosan as well as quantitatively synthesis of the sorbents. Peaks at 1619 cm^−1^, 1621 cm^−1^, and 1600 cm^−1^ with high broadness and sharpness are related to (i) NH of chitosan (ii) C=O of alginate, (iii) C=N from the crosslinking reagent (GA with amines) [51,53,54,55]. These bands are related to the effect of carboxylate functions (1610–1550/1420–1300 cm^−1^) [51] from alginate and NH (1650–1590 cm^−1^) [51] of chitosan, while the intensities of these peaks are controlled by the ratio of alginate, and also related to the effect of GA with amines. Wei et al. study the effect of carboxylate (from methacrylic monomer) on the sorption of Sr, in which the intensity of carboxylate peaks is gradually increased by quantitatively additive of the methacrylic moiety [56]. Again, broad peaks at 3500 cm^−1^, 3399 cm^−1^, and 3197 cm^−1^ of these sorbents respectively are assigned to OH overlapped with NH [57] from the polysaccharide of either chitosan and alginate.

As metal ions were adsorbed, the broadening, resolve of bands, and shifts of peaks were recorded [58]. The mainly observed bands are assigned for OH and NH which shifts from 3399 cm^−1^ and 3419 cm^−1^ to 3364 cm^−1^ with decreasing the intensity (for Pb loaded sorbent), or disappearing of the band (in Cd loaded sorbent). The resolved intensities of N-O and C-O were decreased and shift from 1060 cm^−1^ and 1035 cm^−1^ in the unloaded sorbent to 1026 cm^−1^ and 1033 cm^−1^ for Cd (II) and Pb(II) loaded sorbent respectively. Sharp peaks of C=O (1621 cm^−1^) and COO^−^ (1384 cm^−1^) in the CA#3 were shifted to 1593 cm^−1^ and disappearing of carboxylate was assigned to the peak after Cd adsorption, while after Pb adsorption, these peaks are observed at 1633 cm^−1^ and 1384 cm^−1^ with a decrease in the T%. The shoulder at 615 cm^−1^ (-CH and -OH bending) was completely disappeared with metal sorption.

This indicates the contribution of NH, OH, N-O, and the C=X (N or O) in metal binding sorbent due to the change of the chemical environment of these groups. Additionally, the expected tautomerization of C=O with neighboring atoms assisting the binding of metals and shift/or decreasing the resolution of this peak. The most marked binding was observed by Cd than Pb ions, indicating the efficiency of this metal binding.

After desorption, most of these groups were restored, confirming the chemical stability of these sorbents even after five cycles of sorption desorption. Appendix A (see Appendix A) shows the FTIR spectra for CA#2 and CA#4 before and after sorption and after five cycles of sorption desorption processes. These spectra showed the change of the environments (shifts and disappearing) of some peaks that respond for binding, while the relative stability of these sorbents was shown after five cycles as discussed for CA#3.

#### 3.1.2. Thermogravimetric Analysis and Textural Properties (N_2_ Adsorption)

Different profiles of decomposition plateaus were shown for the three sorbents. The total loss was recorded to around (60–65%) as shown in Appendix A (see Appendix A). This means that the remains are mainly carbon and calcium ions from the organic matrices and ionotropic gelation, respectively. CA#3 and CA#4 show relatively identical weight loss profiles in the first stage attained to the surface and internal water loss (12.8–14.5% respectively) which is assigned at 173.58 °C (18.62–173.58 °C) and 195.78 °C (43.08–195.78 °C), respectively. For CA#2, a different profile is detected, it shows more splitting marked profiles. Two stages were found (first at 100.56 °C (42.47–100.56 °C) and the second at 190.69 °C (100.56–190.69 °C)) with a more extensive loss (22.72%). The next step is the same profile for all, the maximum rate was observed at 293.8 °C, 298.7 °C, and 297.9 °C for CA#2, CA#3, and CA#4 respectively with loss percent around 21.34%, 30.51%, and 25% respectively. This stage is assigned to depolymerization, cleavage of the crosslinking bonds (i.e., around 120 °C), and degradation of functional groups (i.e., amines and hydroxyls)(around 180 °C) [59,60].

The final step of decomposition shows a relatively similar profile. The loss at this stage is recorded to 15.64%, 21.44%, and 21.97% for CA#2, CA#3, and CA#4, respectively. The temperature of this stage was in the range of (293.85–690.84), (298.78–694.08), and (297.96–690.29) respectively. This is related to char decomposition. Different waves were observed from dTGA, Appendix A (see Appendix A). Further, 3, 5, and 4 waves were observed for CA#2, CA#3, and CA#4, respectively. Sharp wave at 99.98 °C for CA#2 compare with low intensity for the others (at 71.54 °C and 96.63 °C for CA#3 and 4 respectively). These peaks are related to releasing of water sorbed sorbent. Peaks at 190.69 °C for CA#2, 173.59 °C and 205.68 °C for CA#3 and 206.05 °C for CA#4 are assigned to cleavage of the crosslinking types and depolymerization process. Peaks at 293.35 °C, 292.56 °C (strong), and 297.96 °C were assigned to degradation of functional groups and polymer chain. CA#2 seems to be completely degradable after this stage. While others show smaller peaks at 422.32 °C and 408.46 °C for CA#3 and CA#4 respectively, these are assigned mainly to the decomposition of the polymer and emphasizing the relation of the thermal stability with alginate additives.

The surface area of CA#2, CA#3, and CA#4 are reached around 39.765, 45.0725, and 46.546 m^2^ g^−1^ respectively, while the pore volume is reached around 13.6 cm^3^ g^−1^, 19.05 cm^3^ g^−1^, and 20.13 cm^3^ g^−1^ respectively, indicating the additive of alginate improve the pore size and consequently the pore volume especially by comparison of 1:2 and 1:3 while 1:4 do not effective so much comparing to 1:3 ratio. The high surface area of the polymer is the main reason for the high sorption and fast kinetics.

#### 3.1.3. Determination of pH_PZC_

The pH_PZC_ of chitosan with different incorporation ratios of alginate shows a remarkable difference. The pH_pzc_ is close to 6.614, 4.88, and 4.42 for CA#2, CA#3, and CA#4 respectively, as shown in Appendix A (see Appendix A). Various functional groups used for controlling the behavior of the polymer, i.e., amines from chitosan, with pK_a_ close to 4.5, 6.7, and 11.6 for 1°, 2°, and 3° amines respectively, this is the main factor for increasing the pH_pzc_ values (support the alkalinity), the function as carboxylic from alginate (two kinds of carboxylic acid; mannuronic (pK_a_, 3.38) and guluronic acid (pK_a_, 3.65)) contributes decreasing this ratio (acidic character). Hamza et al. [53] report the variety in the pH_pzc_ values is mainly depending on the grafted or modified additive functional groups. From these results, it was shown that the sorbents below the pH_pzc_ values are partial to completely protonated. The capacity of loading is depending on the species of metal ions and the pH of the solution, which may bound by electrostatic attraction/or ionic exchange on the partially protonated atoms. While after these pH values, the sorbents become completely deprotonated and bounded with positively charged atoms by non-electrostatic attraction.

#### 3.1.4. XPS Characterization

XPS analysis was used to confirm (i) the chemical modifications of the synthesized material and (ii) investigate the type of bonding for the loaded sorbent through changes in the chemical environment of atoms after metal sorption (i.e., Cd(II) and Pb(II)). The results of deconvoluting of the main peaks and shifts after sorption were summarized in Appendix A (see Appendix A). Figure 2 shows the XPS survey spectra of CA#3 (the most effective sorbent), and after loading with Cd and Pb metal ions. The prepared hydrogel was characterized by C 1s, O, (1s, 2s, OKL1 and OKL2 (weak)), N 1s, and Ca 2p. The loaded sorbent is characterized by the disappearance of Ca 2p, which confirms use in the ion exchange process with Cd^2+^ and Pb^2+^ from the solution. Cd^2+^ ion was confirmed by Cd (4s, 3d3, 3d5, 3p1 and 3s), while Pb found in the loaded sorbent as Pb (3d3, 3d5, 4f7, 4f5, and 4p3). Other ions are appeared in the loaded sorbent as Cl 2p either from medium (chloride medium) or metal-binding chloride species; which indicated that used in the sorption process. The high-resolution spectra (HRES) of some selected bands (as well as their deconvolution) with the assignment of the FWHM and BE were reported in Appendix A (see Appendix A).

A remarkable change of the C 1s is demonstrated in the loading of metal ions. The increase in the FWHM for C (=O, -O, N_tert_) comparing with the raw material from 1.52 eV to 1.98 eV and 1.72 eV for raw composite, Pb and Cd loaded sorbents respectively, with increasing the At% from 8.06% (for unloaded sorbents) to 37.9% and 20.28% respectively, indicated using of these groups in the binding mechanisms. This is accompanied decreasing of C (C, H, N) peak from 1.67 eV (FWHM) and 44.56% (At%) on the unloaded material to 0.86 eV (31.14%) and 1.1 eV (22.81%) respectively. This confirms using of primary and secondary amines for binding with metal ions. Four deconvoluting peaks were observed for the unloaded sorbent for C=C, C (C, N), C-(O, =C), and C(=O, O-C, and N_tert_) at 283.93 eV, 284.35 eV, 285.82 eV, and 286.81 eV respectively. The C=C peak was disappeared for Pb loaded sorbent, while others remarkably appeared at 284.17 eV, 285.14 eV, and 287.05 eV respectively, this is confirming the expected tautomerization of C=C with carbonyl and amid groups [61,62].

The N 1s deconvolution shows an increasing number of the splitting peaks, this is related to the metal binding effect. The unloaded sorbent was deconvoluted into two peaks for N(C, =C, H) and N_tert_ at 398.64 eV and 400.66 eV respectively. As metal-binding proceed, new peaks have appeared as well as shifts of the original peaks were observed, i.e, 398.68 eV and 401.06 eV for N(C, =C, H) and N_tert_ respectively with the formation of the other two peaks at 404.57 eV and 405.59 eV which assigned to O^−^N^+^, N(Pb), and NO_2_ respectively. Again, shifts on Cd loaded sorbent with a new peak for N^+^(Cd) has appeared at 400.71 eV [35,63,64]. Displacements of the C=N confirm participation in the binding mechanism which is accompanied by the change of the group environment [65,66].

The O 1s show a remarkable change in the signals as metal sorbed with varies of FWHM and At%. Two deconvoluting peaks were observed for the CA#3 (i.e., C=O and O (C, H) at 530.24 eV and 531.51 eV respectively) [61,67]. A new peak was found for Pb-O at 528.47 eV, this confirms direct interaction with Oligand. Changes in the C (C, H) environment were observed by little shifts to 531.5 eV and increasing in the At% to 88.12%. Another peak has appeared at 532.32 eV, which is related to CO (C, OH) with low FWHM and At% (i.e., 0.7 and 3.22% respectively). This behavior, is not the same as in the case of Cd, increasing the FWHM and At% of the C=O signal to 1.5 and 26.83% respectively, and decreasing for O (C, H) to 2.02%, and 60.14% respectively with a new peak at 232.09 eV indicates the high affinity of Cd(II) to O atoms than Pb (II) (Appendix A) [68,69].

Cadmium (II) sorption is confirmed by Cd 3d which splitting to four individual peaks at 405.29 eV and 409.48 eV for Cd 3d_5/2_ while peaks at 412.84 eV and 416.83 eV for Cd 3d_1/2_ [68,70]. This confirms that the O and N ligands are participating in Cd ions sorption. Other BE was detected for Cd 3p at Cd 3p_1/2_ with several internal peaks (i.e., 636.61 eV, 641.01 eV, 644.28 eV and 648.74 eV). Additionally, the Cd 4d has two peaks at 4.43 eV and 8.95 eV. These results confirm the high tendency of Cd for N and O ligands. Pb 4f was deconvoluted into two peaks at 143 eV and 138 eV for Pb 4f_5/2_ and Pb 4f_7/2_ respectively [71], while Pb 5d appeared at 18.23 eV confirming the binding with O and N atoms.

#### 3.1.5. Elemental Analysis

C, N, O, and H% were analyzed for the prepared sorbents. As expected, the alginate moiety increases of the O and C contents compared with the N fraction from chitosan. The N% decreased from 4.52 mmol g^−1^ to 3.73 mmol g^−1^ and 2.99 mmol g^−1^ for CA#2, CA#3, and CA#4 respectively, Appendix A (see Appendix A). On the other hand, increasing the mole fraction of O for CA#2 to CA#4 indicating the efficient quantitative synthesis by alginate (source of carboxylic).

#### 3.1.6. SEM-EDX Analysis

Appendix A (see Appendix A) shows the SEM-EDX analysis of the synthesized sorbents. From the SEM pictures, it was shown a dense and little porous surface structure. The surfaces become denser as the ratio of alginate increases and progressively become a more porous structure. These results were identical with the surface area (N_2_ adsorption) information. This is an indication of the successive modifications which contribute to increasing the heterogeneities properties through polarities and affected the porous network. From the EDX analysis, the decrease in the N content (from chitosan) from 5.6 mmol g^−1^ to 4.8 mmol g^−1^ and 2.9 mmol g^−1^ for CA#2, CA#3, and CA#4 respectively, with increasing of O and C, confirms the successive modification and designing. By comparing the molar fraction of N and O from elemental analysis and the EDX analysis, a little increase of the EDX analysis for the CA#2 and CA#3, while the CA#4. On the other hand, increasing the Ca% with alginate ratio gives evidence for the successive ionotropic gelation of the carboxylic groups.

### 3.2. Sorption Properties

#### 3.2.1. pH Effect

Figure 3 identifies the pH effect of Cd(II) and Pb(II) sorption capacities on CA#2, CA#3, and CA#4 sorbents. As pH increased, the amines and hydroxyl groups are progressively deprotonations, causing reducing the repulsion effect of cationic metal species (from the solution) and protons (from the sorbent), which causes an improvement of the sorption efficiency. It was shown that Pb(II) and Cd (II) follow the same behavior until equilibrium. The pH of equilibrium (pH_eq_) is close to 4 for the three sorbents (pH_in_ = 5). The sorbents show different sorption capacities depending on the alginate ratio. CA#3 shows the highest sorption capacities for both elements, the mean average of repeated experiments are closed to (92.43 mg Cd g^−1^/ 0.822 mmol Cd g^−1^, and 87.02 mg Pb g^−1^/ 0.42 mmol Pb g^−1^), compared to other sorbents (i.e., CA#2; 0.6 mmol Cd g^−1^ and 0.323 mmol Pb g^−1^ while CA#4; 0.627 mmol Cd g^−1^ and 0.37 mmol Pb g^−1^). This indicates the efficient addition of alginate to chitosan and its relation to enhancing the sorption capacity. The presence of carboxylic groups supports the chelation properties of the sorbent toward positively charged metal ions. As pH increased above 5, the sorption capacity slightly increased.

Appendix A (see Appendix A) shows the variation of pH during sorption. It has a little difference (i.e., ΔpH around 0.4 unit). The sorption shows three sections (a) acidic (i.e., pH 1), the relative limit of sorption (below 0.18 mmol Cd g^−1^ and 0.03 mmol Pb g^−1^), at this pH, the sorbent is strongly protonated (low pH values), (b) in the range of pH 2–5; steep increase in the loading properties to pH 5 (as pH increase the amine and hydroxyl groups become deprotonated) for enhancing the sorption capacity, (c) the relative stabilization part after pH 5 with little or neglected sorption of both metals. The difference in the sorption of the two elements may regard the radius of hydrated species for each metal ion or softness. The ionic radius (Å) of hydrated species for Cd (II) and Pb(II) is around 0.95–0.96 Å [72] and 1.20 Å [73] respectively, while the softness of both metals ranged about +0.58 and +0.41 respectively. Other details about the diffusivity in water were shown in Appendix A [74]. Appendix A (see Appendix A) shows the SEM-EDX analysis of the sorbent (CA#3) after sorption by Cd(II) and Pb(II) at pH 5.

Cadmium and lead show relative stability with chloride form, which presents in the acidic solution as neutral or cationic species. This is the main reason for decreasing the sorption in the acidic medium (mainly MCl^+^, (i.e., M = Cd or Pb)), repulsion of the positively charged metal ions with protonated groups on the sorbent has happened. As pH increased, the possibility of deprotonation of amines and hydroxyl groups become increases, and the possibility of electron pairs to bound with the positively charged ions (i.e., Cd^2+^ and Pb^2+^ (predominate) or CdCl^+^ and PbCl^+^ (minor)) through chelation was performed. This gives evidence for the presence of chloride ions in the EDX and XPS analyses. The metal ions may bind as M^2+^ or in chloride form.

#### 3.2.2. Sorption Kinetics

The uptake kinetics were investigated toward Cd(II) and Pb(II) sorption. The experimental condition was set to (C_0_: 100 (±5) mg L^−1^; SD: 250 mg L^−1^, and pH_0_: 5). The sorption is fast, depending on the sorbent kind and the metal ions used. One-hour contacts are sufficient for Cd sorption on CA#2 and CA#4, while CA#3 needs 50 min contact for saturation. Sorption passed through two phases, (a) sorption on the surface through the external functional groups, this kind of sorption was characterized by fast and instantaneous, (b) sorption through the reactive groups in the internal parts through the pores. For Pb ion, the saturation time seems to be close with Cd for CA#2 and CA#3, while CA#4 required more time for around 90 min. This is due to the effect of amines and carboxylic groups that grafted on the sorbent. The mass transfer was expected for this sorbent due to micron size. Otherwise, we expect a reduction of the intraparticle diffusion, and the sorption was controlled by resistances to diffusion (through film, and bulk) as well as the rate of reaction. Different reaction rates were detected for kinetics, PFORE and PSORE models [45] fit the kinetic profiles (Appendix A, see Appendix A). Figure 4 and Table 1 compare the PFORE and PSORE modeling profiles, Appendix A (see Appendix A). The PFORE fits the sorption profiles for both elements. The closer values of calculated and experimental results and the correlation coefficients are the main parameters for the priority of one profile over the other.

#### 3.2.3. Sorption Isotherms

Sorption isotherms for the three sorbents were measured at pH_0_ 5 (Figure 5 and Appendix A, see Appendix A). The sorbents show a steep initial slope before equilibrium plateau for both elements, in which CA#3 is steeper than CA#4 and CA#2. The residual metal concentration for both Cd(II) and Pb(II) is close to 2.5 (±0.4) mmol Cd L^−1^ and 1.5 (±0.3) mmol Pb L^−1^ respectively. On the other hand, the sorption capacity (maximum) is 1.42, 1.89, and 1.61 mmol Cd L^−1^ for CA#2, CA#3, and CA#4 respectively, while close to 0.75, 1.054, and 0.87 mmol Pb L^−1^ respectively. The difference in the sorption for both elements may be related to Pearson’s rules [76], hard and soft (H&S) acid–base theory. Yang and Alexandratos [77] reported about a series of different functionalized sorbents grafted by donor atoms for lanthanides extraction based on Soft Acid and Base theory. Other parameters that influence the interaction of ligands and metal ions were investigated, this includes the coordination effect, ligand protonation, and hydration. The polarizability of Cd(II) and Pb(II) [74] makes preferential binding to N and O donor atoms. Giraldo et al. [34] show the high sorption of lead by gelatin activated carbon sorbent beads over Zn(II), Cd(II), and Cu(II) metal ions which are related to the ionic radius. Polystyrene grafted by thiourea is used for investigating the sorption affinity and shows high loading capacity for Pb(II) than Cu(II) and Cd(II). The difference in the hydrated radius [78] of Cd and Pb explains the difference in the sorption affinity of Cd over Pb (i.e., 0.9 Å, and 1.20 Å for Cd(II), and Pb(II) respectively), see Appendix A. Figure 5 shows the most fitted model toward Cd and Pb (Langmuir and Sips). The parameters of each model (Langmuir, Freundlich, and Sips) are reported in Table 2. By comparing the theoretical capacity and determination coefficients (R^2^), it was shown that the Freundlich model (power-type) is the worse fit model for both elements, Appendix A (difficult matching of both experimental data and fitted curves profile). On the other hand Cd(II) and Pb(II) for the three sorbents were fitted with Langmuir (mechanistic equation) and Sips models, (the Langmuir is more close for Cd(II)) while the Sips equation fits the Pb(II) sorption isotherm closer than Langmuir. The q_m_ of the monolayer saturation for Pb(II) is overestimated than that for Cd(II) of the experimental values. Additionally, the affinity coefficients (b_L_) are assigned to be the highest initial slope for Cd than Pb, this is not the same as that of the Sips equation (Table 2).

Table 3 reports a comparison of sorption capacities of different sorbents for Cd(II) and Pb(II). From the first point of view, the synthesized sorbents have a high capacity and relatively fast sorption kinetics to cite this sorbent as one of the most efficient for Cd(II) and Pb(II). Some sorbents show a relatively high capacity than CA#X (x = 2, 3, and 4) sorbents, as cationic resin (001 × 7), and HAHZ-MG-CH sorbent, while the sorption kinetic (i.e., 120 min and 60 min) of these sorbents are saturated in much longer time than the CA#X sorbents (maximum saturation time less than 50 min). Mercaptoamine on silica-coated magnetic nano-sorbent, gelatin/activated on carbon beads, tripolyphosphate functionalized chitosan and HA-MG-CH showed high capacity than the CA#X toward Pb(II), but as discussed on the Cd, the main advantage of this sorbent concerns the saturation time (around 40–50 min) while the other sorbents were saturated in around (180 to 60 min).

#### 3.2.4. Expected Binding Scheme

We summarize the data mentioned in the XPS survey, FTIR analysis, the effect of sorption as a function of pH, the pH_PZC_, and the semi-quantitative EDX analyses. We expected different interaction modes of the reactive groups present on the designed hydrogel and the metal ions in the solution. The material consists of multi-functional groups as amines (1°, 2°, and 3° amines) from chitosan moiety, hydroxyl groups from the polysaccharide of chitosan and alginate, and carboxylic groups from alginate. These groups are symmetrically arranged for binding with the metal ions. The functional groups in the sorbent are completely deprotonated (see the pH_pzc_ analysis). Participating are the amines and hydroxyls (decreasing the intensities of OH and NH in the FTIR), tautomerization of C=O with the neighboring groups for bending (decreasing and shifts of the assigned peak in the FTIR and XPS analyses), also chelation with carboxylic groups as well as ionic exchange mechanism with the Ca^2+^ ions from the ionotropic gelation (disappearing of Ca^2+^ ions from the loaded sorbent in the XPS analysis, and also shifts of the C=O and COO^−^ of the carboxylate salts in the FTIR analysis). The metal ions are found in the form of free metal cations (M^2+^) or as monovalent chloride form (as assigned from the XPS analysis), as summarized in Scheme 2.

#### 3.2.5. Metal Desorption and Sorbent Recycling

Appendix A (see Appendix A) shows the comparable desorption properties of the synthesized sorbents toward Cd(II) and Pb(II) using 0.2 M HCl. Different behavior of desorption was observed, i.e., the time that was taken for complete desorption. This eluent reagent is sufficient for completely eluting the adsorbed metal ions.

Desorption seems faster than the adsorption profile; 15–40 min is sufficient for complete desorption. CA#3 is the slowest sorbent for releasing the adsorbed metal ion, also Pb(II) is more efficient than Cd(II) (40 min for Cd(II) and 30 min for Pb(II) for complete desorption). The other sorbents required shorter time, i.e., 15 and 30 min was sufficient for complete desorption of Cd from CA#2 and CA#4 respectively. On the other hand, 30 min for Pb(II) appeared to be sufficient for completely desorbed metal ions.

The most efficient sorbent for either Cd and Pb is CA#3, which leads to further experiments concerning sorption desorption cycles (stability). Table 4 reports the cycles of sorption and desorption. The sorbent shows stability for five cycles, the loss in the sorption does not exceed 4% and around 5% for Cd and Pb respectively, while desorption remains stable after the five cycles (or little decrease in the efficiency, up to 0.1% and 0.01% respectively). The chemical stability was detected by FTIR analysis (Figure 1, Section 3.1.1) which exhibits restoring of the function groups after five cycles.

#### 3.2.6. Sorption Characteristic (Multi-Component Solution)

The sorption from equimolar multi-component ions was investigated for CA#3 at different pH values. The choice of metal ions is based on the mainly presented in the contaminated waste samples, especially that of the high salinity solution, among these ions, Ca(II), Mg(II), and Al(III). This study focused on the selectivity of the grafted groups (amine, amide, hydroxyl, and carboxylic groups) for Pb and Cd in comparison with other associates. The grafted groups give broad flexibility for metal ions to bind through chelation, especially at a low acidic solution as pH 5. Appendix A (see Appendix A) shows the selectivity coefficients of Cd and Pb ions comparing to the other associates (SC_Cd/metal_ and SC_Pb/metal_). The selectivity coefficients were calculated from the below equation.
(1)$$\mathrm{Selectivitycoefficient}\left(\mathrm{SC}\right)=\frac{D\left(\mathrm{CdorPb}\right)}{\mathrm{Dmetal}}=\frac{\mathrm{qeq}\left(\mathrm{CdorPb}\right)\times \mathrm{Ceq}\left(\mathrm{metal}\right)}{\mathrm{Ceq}\left(\mathrm{CdorPb}\right)\times \mathrm{qeq}\left(\mathrm{metal}\right)}$$

The sorbent preferentially of Cd over Pb in the multi-component solution at a higher pH value similar to the mono component solution experiment. The SC of SC_Cd/metal_ at pH 2.43 ranged from 0.9–1.4 for Cd over other ions, while for Pb, it reached around 0.6 to 1.7. As pH increased, the efficiency toward Cd and Pb becomes preferentially than the associated ions, and the selectivity coefficient accordingly increased. At pH 3.68 to 4.76, the sorbent shows priority for Cd than other elements by around 1.7 to 6.7, 3.1 to 25.7, and 3.5 to 22 times respectively depending on the metal. The lowest selective properties were obtained with Pb and the highest for Al ion. On the other hand, for Pb, the selectivity arranged 0.56 to 3.8, 0.3 to 8.2, 0.2 to 6.3 times respectively. These data are arranged in Appendix A. The highest selectivity was reached at high pH values and arranged according to Cd(II) > Pb(II) > Mg > Ca > Al.

### 3.3. Treatment of Contaminated Water

The Abu Zaabal site is a famous and highly contaminated area, Appendix A (see Appendix A). The collected water sample from Abu Zaabal lake was enriched with Cd and Pb at around 1.96 and 2.03 mg L^−1^ respectively, and other elements (i.e., Zn, Cu, Al, and Fe) were detected. The sorption capacity of CA#3 was controlled as a function of pH (i.e., 5.8 (original pH of the taken sample), 4.04 (the partially protonated sorbent statement), and pH 2 (acidic pH)). The batch method was used for 5 h of contact, while the SD was fixed in all experiments at 1.0 g L^−1^. The sorption capacity and the total recovery for each element were determined using the mass balance equations as discussed before. The sorption efficiency improved with pH (parallel to synthetic solution). The removal efficiency was studied under the experimental condition and found to be around 85.48%, 98.31%, 96.95%, 91.14%, 82.27%, 99.44%, 99.21%, and 95.64% for Cu, Hg, Fe, Al, Zn, Cd, Pb, and Ni respectively from the original pH value (5.8). The sorption efficiency varied depending on the pH and metal ions. Appendix A reports the sorption capacity of each element at different pH values, the original, as well as the residual (after treatment). The residual concentrations are compared with the MCL (maximum contaminant level) for the drinking water assignments according to the World Health Organization [1].

Some concentrations are still higher than the allowance for drinking water. Zinc(II) has a high concentration in the MCL (5 mg L^−1^) and a lower concentration in the effluent (1.31 mg Zn L^−1^). The high level for the livestock feeds is 0.05 mg Cd L^−1^, and 0.1 mg Pb L^−1^, and 24 mg Zn L^−1^. Appendix A (see Appendix A) shows the selectivity coefficient of Cd and Pb comparing to the other elements at the selected pH values. It was shown that as pH increased, the loading capacity toward metal ions increased, showing the main improvements achieved with Cd and Pb. The SC_Cd/metal_ at pH 5.8 reached 3.04, 8.07, 30.11, 5.58, 38.18, 17.23, 8471.51, and 1.41 times for Hg, Ni, Cu, Fe, Zn, Al, Na, and Pb respectively, while the SC_Pb/metal_ at the same condition of pH value is about 5.74, 21.39, 3.96, 12.24, 6018.4, and 27.12 times respectively while reached around 0.71 for Cd. At the neutral pH, the selected experimental conditions are suitable for the livestock feed. In all cases, the final concentration can systematically be improved by increasing the SD. Data represented in Figure 6 show comparable studies of the removal efficiencies after treatment of water samples at different pH values. The maximum removal of metal ions was obtained at high pH values of 5.8. The R% for the metal ions at pH 5.8 is about 85.48, 98.31, 96.95, 91.14, 82.27, 99.44, 99.21, and 95.64% for Cu, Hg, Fe, Al, Zn, Cd, Pb, and Ni respectively, and these ratios decreased with decreasing the pH value to 69.08, 76.63, 93.38, 71.46, 54.67, 86.73, 72.41, and 50.92% respectively for pH_in_ equivalent to 4. The poor recovery was obtained for the acidic pH value that reached 20.04, 53.74, 55.87, 56.44, 6.49, 7.65, 9.85, and 4.54% respectively, in which Hg, Fe, and Al show the highest R% (around 55%) over the others (less than 10% except for Cu around 20%).

## 4. Conclusions

A newly designed biopolymer member family of chitosan was prepared for enhancing the sorption efficiency. The synthesized polymer was crosslinked by GA and ionotropic gelation was done by CaCl_2_. The prepared sorbents were synthesized by different ratios of alginate to the fixed amount of the chitosan followed by crosslinking and ionotropic gelation. Sorbents CA#2, CA#3, and CA#4 of 1:2, 1:3, and 1:4 of chitosan and alginate respectively were synthesized. These eco-friendly and cost-effective sorbents were characterized by different analytical tools; physical characterization (i.e., SEM, TGA, and N_2_ adsorption), chemical characterization (FTIR, EDX, titration, and XPS). The sorbents have various reactive groups (hydroxyls, amines (from chitosan), carboxylic (from alginate), amide and carbonyl groups (from the crosslinking agent)), and all of these groups combine in good symmetry to enhance the sorption efficiency toward Cd(II) and Pb(II) as well as improve the stability (no noticeable decreasing in the efficiency after five cycles of sorption desorption). The optimum sorption of the different synthesized sorbents was recorded at pH_0_ 5. The mean average sorption capacity for CA#2, CA#3, and CA#4 is closed to 0.6 mmol Cd g^−1^, 0.822 mmol Cd g^−1^, and 0.627 mmol Cd g^−1^, respectively, as well as 0.323 mmol Pb g^−1^, 0.42 mmol Pb g^−1^, 0.37 mmol Pb g^−1^ respectively. More than 99% of adsorbed ions were desorbed by 0.2M HCl with contact for 15 to 30 min. Loss of sorption capacity was observed for around 4% from Cd and 6% from Pb by five cycles of sorption desorption, while a negligible loss in desorption performances was obtained. Langmuir and Sips’ equations fit the isotherm profile of both elements more than the Freundlich equation. However, PFORE fits the kinetic profiles for both elements.

The sorbent (CA#3) was tested toward the multi-component solution of mono, di, and trivalent metals and shows high affinity toward Cd and Pb especially at high pH values. At the final step of this study, the sorbent (CA#3) is subjected to the removal of contaminant from a real water sample in the industrial area collected from Egypt. The experiments were performed at different pH values and compared to the final concentration of the water samples produced after sorption by the MCL according to WHO and another organizer, also by the maximum levels of the contaminants for livestock and irrigation of water. The prepared sorbent shows high recovering of the contaminant even in a solution with low concentrated metal ion as that found in the underground water and industrial samples that applied for the water treatment technology.

## Data Availability

The data presented in this study are available on request from the corresponding author.

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
