# Peer review of "Synthesis of Eco-Friendly Biopolymer, Alginate-Chitosan Composite to Adsorb the Heavy Metals, Cd(II) and Pb(II) from Contaminated Effluents"

_materials, 2021, doi:10.3390/ma14092189_

Round 1

Reviewer 1 Report

The paper presents an attempt to describe the synthesis of eco-friendly biopolymer, alginate-chitosan composites and the mechanism of adsorption of the heavy metals, cadmium and lead from contaminated effluents. In my opinion the presentation of the investigation methods as well as the scientific results are not satisfactory for the paper to be recommended for publication. The minor and major drawbacks to be addressed can be specified as follows:
1.    Page 1. (i) redundant comma: “Adel A-H., Abdel-Rahman,” ---> “Adel A-H. Abdel-Rahman,”, (ii)no dot: “Ibrahim F Zeid” ---> “Ibrahim F. Zeid”, and (iii) redundant dot: “Salem7.” ---> “Salem7”.
2.    Page 1, line 28. BET is the theory or the method: “BET” ---> “nitrogen adsorption (T=77K)”. See also line 95.
3.    Page 1, lines 40 and 41. “cadmium, and lead contamination” ---> “cadmium and lead contamination”.
4.    Pages 2 (line 99) and 4 (line 146). Why only twice? The typical number of measurements is three repetitions.
5.    Page 3, line 117. w/v ---> W/V. See line 113.
6.    Page 3, line 121. scheme ---> Scheme.
7.    Page 3, Scheme 1. Why do the colors appear in this scheme on the right and what do they mean? Please explain it (figure captions) and introduce colors on the left and middle parts of this picture.
8.    Page 4, line 136. BET properties ---> Adsorption properties.
9.    Page 4, line 138. The samples should be degassed at a temperature higher than 100°C, for example 105-120°C. This is my suggestion for the future.
10.    How was the calcium content of the samples experimentally estimated?
11.    Page 4, line 163. Modeling ---> Describing.
12.    Page 4, lines 169- 177. References?
13.    Please standardize: Page 4, line 163, “S.D.”? Page 1, line 38, “SD.”? Page 10, line 393, “SD”.
14.    Page 4, line 154. “sorbent dosage” ---> “sorbent dosage (S.D.)” or “sorbent dosage (SD.)” or “sorbent dosage (SD)”. See above.
15.    Page 5, line 192, “CA#2, CA#3, and CA#4”. Please explain it (introduce the respective markings/abbreviations) for the first time on Page 3, lines 115 and 116. (i.e., my proposition – “for 1:2, 1:3, and 1:4 of Chitosan:Alginate, respectively, so-called CA#2, CA#3, and CA#4”).
16.    Page 5, Fig. 1. (i) A and B are hardly visible. In my opinion the best position of the description of curves (related to the respective samples) is area outside the panel on the right.
17.    Page 6, lines 218-227. Discussion with other results? References?
18.    Page 6, line 259 and others. Is it necessary to specify the values of temperature with such a high degree of accuracy?
19.    Page 7, lines 267-270. Is it necessary to specify the values of the apparent surface area with such a high degree of accuracy? Please add to the Supplementary Materials the new figure with the low-temperature nitrogen adsorption/desorption isotherms. Please estimate the total volume of the available pores from adsorption isotherms.
20.    Page 10, Fig. 3. I would avoid connecting the points with a line.
21.    Page 16, lines 403 and 404, “to the radius of hydrated species for each metal ion or softness”. Please collect the respective values taken from the literature in the manuscript.
22.    Page 12, Tab. 1, table captions. Please explain “EV” and “AIC” (AIC should be referred to the equation form the Supplementary Materials, lines S55 and S56).
23.    Page 11, lines 433 and 435, “The closer values of calculated and experimental results and the correlation coefficients are the main parameters for the priority of one profile than the other.” See Tab. 1. Correlation coefficients? What about R2, EV, and AIC? Are these quantities and values (collecting in this tables) discussed?
24.    Page 13, Tab. 2. Why EV was not calculated?
25.    Page 13, Fig. 5, legends. My proposition: two columns in the legend, 1st: CA#2Cd, CA#3Cd, and CA#4Cd; 2nd CA#2Pb, CA#3Pb, and CA#4Pb.
26.    Page 14, Tab. 3. The following part of this table is incomprehensible. 
                              Equilibrium    qm(mmo
                              time (min)    l g–1
        The text is rearranged. The parenthesis is not closed. My proposition
                              Equilib. time          qm
                                   (min)         (mmol g–1)
27.    Page 16, Tab. 4. (i) Table captions: “Cycles of Sorption” ---> “Cycles of sorption” (ii) Why negligible? Please give the respective values.
28.    Pages 16 and 16, Eq. 1, Fig. 6. Where did the authors get the Ceq(metal) and qeq(metal) values for other metals?
29.    Page S2, Tab. S1. “mg/L” ---> “mg L-1”.
30.    Page S3, Tab. S3. (Foo and Hameed, 2010; Tien, 1994)????? See line 166 and references [46,47], i.e. (Foo and Hameed, 2010; Freundlich, 1906).
31.    Page S4, Tab. S4. For example, “C0/MCL ratio = 551.33333”, too high degree of accuracy?
32.    Page S5, Tab. S5. For example, for example, CA#3+Pb or CA#3Pb (Fig. 5).
33.    I do not check the Supplementary Materials (from Tab. S5) due to a large number of minor errors. Authors must review the manuscript very carefully prior to the further review.

Author Response

Thank you very much for reviewing our manuscript. We made a point-by-point response to the reviewer’s comments, and we hope they meet with your approval.

Reviewer 2 Report

Authors performed an extensive research on heavy metal removal by chitosan based composite materials. The study is interesting and relevant, however the language is very bad, and needs an extensive revision.

Experimental:

line 136  "BET properties" should be replaced by pore morphology or porosity or something similar.  "were used by"  -- should be replaced by grammatically appropriate expression.

The temperature of the sorption experiments, and the instruments used should be explained in more details.

Table 1 caption should be Cd and Pb

Table 1 and 2:  indication of units of   mmol M g–1  is somewhat non standard, the "M" can be omitted. 

Table 3: comparison of Cd and Pb adsorption on the widely used inorganic mesoporous silica based materials can also be included. For example:

mdpi.com/2076-3417/10/8/2726/htm

https://doi.org/10.1039/C9ME00140A

https://doi.org/10.1016/j.jallcom.2009.11.204  

Figure 3 bottom, left axis: should be Pb instead of Pr  

Some grammar corrections are necessary:  

abstract

Langmuir and Sips are well fitted the isotherms over Freundlich model. may be rewritten as: Langmuir and Sips models fitted better the adsorption isotherms compared to the Freundlich model.

sentence in lines 562-563 should be rewritten clearly. That whole paragraph can also be rewritten.

Some examples where the text must be improved:

line 221 "The resolve intensities"

line 267 " is reached"

line 268 "improce"    

line 270 "has the main reason" -- should be "is"

line 484 “ … is overestimated than that ... The data availability statement should be provided. 

line 601 and figure 6 caption: should be comparative instead of comparable

References

in general, the journal full names, or abbreviated names should be used, in a standard way. Many of the references need corrections.  

ref.2. the article id number should be given.

ref 3. doi is typed twice ref 12 journal name is written incorrectly

ref 49, the doi and the page numbers are not separated, DOI is not indicated. missing doi numbers can be added, e.g. to refs 39, 44

Author Response

(The authors gave the same response as above.)

Round 2

Reviewer 1 Report

The authors have made a substantial improvement for this article. The manuscript can be accepted for publishment in the present form.